# Dolutegravir-Based Regimen Ensures High Virological Success despite Prior Exposure to Efavirenz-Based First-LINE ART in Cameroon: An Evidence of a Successful Transition Model

**DOI:** 10.3390/v15010018

**Published:** 2022-12-21

**Authors:** Ezechiel Ngoufack Jagni Semengue, Joseph Fokam, Naomi-Karell Etame, Evariste Molimbou, Collins Ambe Chenwi, Désiré Takou, Leonella Mossiang, Alain P. Meledie, Bouba Yagai, Alex Durand Nka, Beatrice Dambaya, Georges Teto, Aude Christelle Ka’e, Grâce Angong Beloumou, Sandrine Claire Djupsa Ndjeyep, Aissatou Abba, Aurelie Minelle Ngueko Kengni, Michel Carlos Tommo Tchouaket, Nounouce Pamen Bouba, Serge-Clotaire Billong, Samuel Martin Sosso, Vittorio Colizzi, Carlo-Federico Perno, Charles Kouanfack, Anne-Cecile Zoung-Kanyi Bissek, Emmanuel Eben-Moussi, Maria Mercedes Santoro, Francesca Ceccherini-Silberstein, Alexis Ndjolo

**Affiliations:** 1Chantal BIYA International Reference Centre for Research on HIV/AIDS Management and Care, Messa, Yaoundé P.O. Box 3077, Cameroon; 2Department of Experimental Medicine, Faculty of Medicine and Surgery, University of Rome “Tor Vergata”, Via Montpellier 1, 00133 Rome, Italy; 3Faculty of Sciences and Technologies, Evangelical University of Cameroon, Bandjoun P.O. Box 0127, Cameroon; 4Faculty of Health Sciences, University of Buea, Buea P.O. Box 063, Cameroon; 5National HIV Drug Resistance Working Group, Ministry of Public Health, Yaoundé III P.O. Box 3038, Cameroon; 6School of Health Sciences, Catholic University of Central Africa, Yaoundé P.O. Box 11628, Cameroon; 7Central Hospital of Yaoundé, Yaoundé P.O. Box 047, Cameroon; 8General Hospital of Douala, Douala P.O. Box 4856, Cameroon; 9PhD Courses in Microbiology, Immunology, Infectious Diseases and Transplants (MIMIT), University of Rome “Tor Vergata”, Via Montpellier 1, 00133 Rome, Italy; 10Directorate for Disease Epidemic and Pandemic Control, Ministry of Public Health, Yaoundé III P.O. Box 3038, Cameroon; 11Central Technical Group, National AIDS Control Committee, Yaoundé II P.O. Box 2005, Cameroon; 12Faculty of Medicine and Biomedical Sciences, University of Yaoundé I, Yaoundé P.O. Box 1364, Cameroon; 13Bambino Gesu’ Children’s Hospital, Piazza S. Onofrio 4, 00165 Rome, Italy; 14Faculty of Medicine and Pharmaceutical Sciences, University de Dschang, Dschang P.O. Box 067, Cameroon; 15Division of Operational Health Research, Ministry of Public Health, Yaoundé III P.O. Box 3038, Cameroon

**Keywords:** HIV, antiretrovirals, first-line, TLD, virological response, Cameroon

## Abstract

To ensure optimal prescribing practices in the dolutegravir-era in Cameroon, we compared first-line virological response (VR) under tenofovir + lamivudine + dolutegravir (TLD) according to prior exposure to tenofovir + lamivudine + efavirenz (TLE). A facility-based survey was conducted among patients initiating antiretroviral therapy (ART) with TLD (I-TLD) versus those transitioning from TLE to TLD (T-TLD). HIV viral load was performed and unsuppressed participants (VL > 1000 copies/mL) had genotyping performed by Sanger sequencing. Of the 12,093 patients followed, 310 (mean-age: 41 ± 11 years; 52.26% female) complied with study criteria (171 I-TLD vs. 139 T-TLD). The median ART-duration was 14 (12–17) months among I-TLDs versus 28 (24.5–31) months among T-TLDs (15 (11–19) on TLE and 14 (9–15) on TLD), and 83.15% (148/178) were at WHO clinical stages I/II. The viral suppression rate (<1000 copies/mL) was 96.45%, with 97.08% among I-TLDs versus 95.68% among T-TLDs (*p* = 0.55). VR was similar in I-TLD versus T-TLD at <400 copies/mL (94.15% versus 94.42%) and age, gender, residence, ART-duration, and WHO stages were not associated with VR (*p* > 0.05). Genotyping was successful for 72.7% (8/11), with no major mutations to integrase inhibitors found. VR is optimal under first-line TLD after 14 months, even among TLE-exposed, thus confirming the effectiveness of transitioning from TLE to TLD in similar settings, supported by strong pharmacological potency and genetic barrier of dolutegravir.

## 1. Introduction

In the frame of HIV treatment, the five main goals of antiretroviral therapy (ART) are (i) maximal and durable control of the viral replication, (ii) prevention of HIV transmission, (iii) restoration of immunological function, (iv) reduction of HIV-related mortality, and (v) increased duration and quality of life for people living with HIV [1]. Regarding treatment regimens, first-generation non-nucleoside reverse-transcriptase inhibitors (NNRTIs), namely efavirenz (EFV) and nevirapine (NVP), were the recommended first-line-based regimens, in combination with nucleoside reverse-transcriptase inhibitors (NRTIs) such as lamivudine (3TC) and tenofovir (TDF) [2,3,4,5,6,7]. However, given the low genetic barrier of NNRTIs to HIV drug resistance and the emergence of EFV/NVP pretreatment drug resistance above the threshold of 10% in several resource-limited settings (RLS), the use of NNRTI-sparing regimens has become a high priority [8,9,10]. In effect, people with NNRTI resistance are less likely to achieve viral suppression, more likely to experience virological failure or death, more likely to discontinue treatment, and more likely to acquire new HIV drug-resistant mutations [11]. In this context, transitioning from EFV/NVP-based therapy to dolutegravir (DTG)-based regimens was of paramount importance [10,12,13,14,15,16,17]. Of note, DTG is an integrase strand-transfer inhibitor (INSTI) with a high potency, a high genetic barrier to HIV drug resistance, and is widely available in a fixed-dose combination consisting of tenofovir (TDF) + lamivudine (3TC) + DTG (TLD) [10,18,19,20]. Therefore, TLD is the current preferred regimen for ART initiation in RLS in replacement of tenofovir (TDF) + lamivudine (3TC) + EFV (TLE) [12,21]. ART guidelines in Cameroon follow the WHO public health approach, and consist of two NRTIs and one NNRTI as a preferred first-line regimen before 2020 [3]. The introduction of TLD as a first-line regimen for treatment initiation was officially launched for newly infected patients as of 1 January 2020 [22], and the transition of the formal first-line regimen (like TLE) to TLD was done in health facilities at the discretion of clinicians; current guidelines recommend a switch to second-line regimens consisting of two NRTIs and one ritonavir-boosted protease inhibitor (PI/r) in the event of first-line ART failures. Importantly, many authors have already reported early findings on the effectiveness of DTG-based first-line ART, including modelling studies and clinical trials in sub-Saharan Africa and beyond [13,21,23,24]. However, despite the predicted high effectiveness of TLD, the effect of previous exposure to the same backbone (TDF + 3TC) gives room for possible existing drug-resistant mutants in the current transition from TLE to TLD (specifically with the K65R and M184V mutations known as major DRMs to TDF and 3TC/FTC, respectively). In effect, Inzaule et al. recently reported that high levels of resistance to the NRTI backbone were common in patients failing NNRTI-based first-line ART in RLS [25]. Moreover, accumulation of NRTI-resistance mutations in patients transitioning to TLD would lead to a “functional” monotherapy of DTG [26], which would further enhance rapid emergence of DTG resistance among such patients. Thus, the transition to TLD in patients with prior exposure to TLE may be accompanied by a suboptimal response in RLS and sub-Saharan Africa in particular. Understanding the treatment outcomes among patients on the recently introduced first-line (TLD) regimen would shed light on potential disparities between those with previous exposure to ART and those initiating ART with DTG within the Cameroonian context. To address this gap, and in order to ensure the long-term efficacy of DTG, we evaluated the virological response under TLD according to previous exposure to TLE in first-line patients in Cameroon.

## 2. Materials and Methods

### 2.1. Study Setting and Site 

A facility-based survey was conducted from June through November 2021 among first-line patients living with HIV (PLWH) initiating ART directly with TLD versus those transitioning from TLE to TLD, all followed at the daycare Hospital of the Yaoundé Central Hospital (YCH) and the Douala General Hospital (DGH), the two main hospitals of the capital cities of Cameroon. Laboratory analyses within the frame of the study were performed at the Chantal Biya International Reference Centre for research on HIV/AIDS prevention and management (CIRCB) in Yaoundé, Cameroon. 

The CIRCB is a government institution of the Ministry of Public Health dedicated to HIV research and patient monitoring in several aspects. Among these aspects are (a) HIV early infant diagnosis in the frame of the national PMTCT program, (b) diagnosis of coinfections with HIV, (c) viral load measurement, (d) CD4 and CD8 T lymphocytes counts, (e) biochemical and hematological tests for drug safety, and (f) genotypic HIVDR testing (GRT) at subsidized costs, with quality-control programs conducted in partnership with the Quality Assessment and Standardization of Indicators (QASI) and other international agencies [27].

### 2.2. Study Population 

Eligible participants were HIV-1 infected patients currently under TLD first-line ART, having been on treatment for at least six months and who consented to participate in the study. Through an exhaustive sampling strategy, two groups of patients were recruited: (i) those initiating ART directly with TLD (I-TLD), and (ii) those transitioning from TLE or TLN to TLD (T-TLD). WHO clinical stages were those reported in patients’ medical reports at treatment initiation. Participants excluded were PLWH with a nondetailed treatment history and those who had received prior ART other than TLE or TDF + 3TC + NVP (TLN). Children and adolescents (≤18 years) were not considered for inclusion in this study as per national guidelines (i.e., current recommendations for treatment initiation are not TLD in this target population).

### 2.3. Viral Load Measurement 

Plasma viral load (PVL) at the CIRCB was performed by using the Abbott^®^ m2000rt HIV platform (Abbott Molecular Inc. 1300 E. Touhy Ave. Des Plaines, IL 60018 200680-105, USA) according to manufacturer’s instructions [28]. Briefly, a protocol using 0.6 mL of plasma was used for RNA extraction, followed by a simultaneous amplification and detection on a real-time polymerase chain reaction (RT-PCR). The lower and upper detection thresholds of the assay were <40 and >10,000,000 HIV-1 RNA copies/mL, respectively. Viral suppression was considered as a viral load <1000 copies/mL, as previously defined by WHO guidelines for resource-limited settings [18].

### 2.4. HIV-1 Genotypic Drug-Resistance Testing

HIV-1 genotypic resistance testing (GRT) was performed on both HIV-1 plasma RNA and proviral DNA following an in-house integrase genotyping assay as described and published elsewhere [29]. Briefly, viral nucleic acids were extracted by using the Qiagen protocol with a viral RNA extraction kit for plasma and DNA extraction for buffy-coat samples. For RNA extracts, reverse-transcriptase and PCR were performed by using the following conditions (1 cycle at 50° for 30 min, 1 cycle at 94 °C for 2 min, 40 cycles (95 °C, 30 s; 51 °C, 30 s, 72 °C, 2 min and 30 s); 1 cycle at 72 °C for 10 min, 1 cycle at 4 °C for 30 min, and 1 cycle at 10 °C forever). For DNA extracts, a direct PCR was performed by using the following conditions (1 cycle at 94 °C for 12 min, 40 cycles (95 °C, 30 s, 51 °C, 30 s, 72 °C, 2 min and 30 s), 1 cycle at 72 °C for 10 min, and 1 cycle at 4 °C forever). Gel electrophoresis, purification, and sequencing reaction were performed in a similar manner.

### 2.5. Interpretation of Drug-Resistance Mutations and HIV Subtyping 

Regarding the analysis of drug-resistance mutations (DRMs), sequences obtained after capillary electrophoresis on the 3500 Genetic Analyzer (Applied Biosystems™, California, USA) were assembled and manually edited by using RECall (CDC, Atlanta, GA, USA). They were then analyzed for interpreting DRMs by using Stanford HIVdb v9. All variants at amino acid positions associated with a decreased INSTI susceptibility (i.e., at least a Stanford penalty score ≥ 10) were considered as resistant variants. Samples with resistant mutant or mixture of wild type and mutant at an amino acid position were considered resistant.

For subtyping, sequences were aligned in BioEdit version 7.2.6 (Tom Hall, Raleigh, NC, USA) by using CLUSTAL W, and compared with reference sequences for the major HIV-1 subtypes and circulating recombinant forms (CRFs), available in the Los Alamos database [30]; gaps were then removed from the final alignment. The phylogenetic tree was inferred by using maximum likelihood method on the MEGA software v7.0.26 primarily for subtyping and to ensure that there was no cross-contamination of samples. 

### 2.6. Statistical Analysis

Data were analyzed by using Excel 2016, Epi-info v7. Chi square or Fisher’s exact test was used to describe the associations between variables wherever appropriate, with *p*-values < 0.05 considered statistically significant; logistic regression was used for multivariate multivariable analyses for variables showing a *p*-values ≤ 0.2 from bivariable analyses.

## 3. Results

### 3.1. General Characteristics of Study Population 

#### 3.1.1. Sociodemographic and Basic Clinical Characteristics

Out of 12,093 patients receiving ART at the reference treatment centres (YCH and DGH), 310 had been on a DTG-based first-line ART for at least six months. The participants had a median (IQR) age of 41 (34–49) years. The most represented age group included those of 35–44 years of age (34.19%), followed by those 25–34 years of age (21.29%), those 45–54 years of age (20.32%), those 55–64 years of age (10.65%), those 15–24 years of age (4.52%), those 65–74 years of age (2.26%), and finally those 75–85 years of age (0.32%). There was a nearly equal distribution of male and female participants (52.26% female; 162/310) in the study population and the majority of participants (about 2/3) lived in Yaoundé. Only 178/310 (57.4%) of participants had data on WHO clinical stage at treatment initiation, and the majority of them were in clinical stages I and II (67.41%; 120/178 and 15.73%; 28/178, respectively), as shown in Table 1.

#### 3.1.2. Description of ART History

The overall median ART duration on a first-line regimen was 19 (13–27) months (Table 1). Regarding ART exposure, 171/310 (55.16%) participants initiated ART directly with TLD (I-TLD) and the remaining 139/310 (44.84%), initiated ART with TLE before subsequent transition to TLD (T-TLD). We did not find any participant transitioning from TLN to TLD. The median viremia before transition to TLD-based first-line therapy among T-TLD participants was 3258 (831–59,276) copies/mL (Table 1) and 89.93% (125/139) had a detectable viral load before the transition. We did not have any record of CD4 count in both arms. ART duration among I-TLD was 14 (12–17) months compared to 28 (24.5–31) months (divided as 15 (11–19) months on TLE and 14 (9–15) months on TLD) among T-TLD. Figure 1 details the algorithm used to enroll and track patients.

### 3.2. Virological Response

The overall rate of viral suppression (PVL < 1000 copies/mL) after approximately 14 months was 96.45% (299/310), without any significant disparity (*p* = 0.55) between I-TLD vs. T-TLD, as shown in Figure 2. Importantly, median viral load for TLD’s non-suppression was 2970 (1334–8866) copies/mL, 1262 (1096–2970) copies/mL, and 8605 (3339–16,417) copies/mL, respectively, among I-TLD and T-TLD.

#### 3.2.1. Factors Associated to Virological Response

Following bi- and multivariate analyses, neither age, nor gender, city of residence, WHO clinical stage, or duration on TLD were found to be associated with viral suppression in the present study (Table 2).

#### 3.2.2. HIV-1 Genotyping and Genetic Diversity in the Integrase Coding Region

Among the 11 patients with unsuppressed viremia (PVL ≥ 1000 copies/mL), HIV-1 integrase genotyping was successful for 72.7% (8/11) of the study participants. Two viral strains were identified, 87.5% CRF02_AG (7/8) and 12.5% F2 (1/8). No major mutations to integrase inhibitors were found.

## 4. Discussion

In view of achieving UNAIDS’ target by 2030, 85% of all people living with HIV worldwide knew their status; 88% had access to ART and 92% achieved viral suppression in 2021 [31,32]. In Cameroon specifically, 94.1% of all people living with HIV know their status, 82.7% of them have access to ART, and 94.1% of those on ART with access to viral load testing achieved viral suppression by 2021 [33]. This progress was a result of rapid ART scale-up which has dramatically reduced HIV-related morbidity and mortality even in RLS [18,34,35,36]. However, despite the tremendous successes of ART scale-up, it is very unfortunate that there has been an increase in levels of pretreatment HIVDR among patients initiating ART [9,25]. The burden of HIVDR is significantly associated with poor virological outcomes and with increased mortality and morbidity [9]. Regimens with greater potency and higher genetic barrier to resistance might be essential in this area. Thus, understanding the treatment outcomes among patients on the recently introduced first-line regimen (TLD) would shed light on potential disparities between those with previous exposure to ART and those initiating ART with DTG within the context of countries similar to Cameroon. 

Participants in this study were aged 41 years on average, with nearly equal distribution of male and female participants, which is in line with the national HIV epidemiology [37], even though other studies highlight female vulnerability to HIV infection [38,39,40]. This highlights the representativeness of our findings to the target population at the country level. Even though the information was absent for some participants, the high number of early WHO clinical stages at treatment initiation translates into overall good clinical condition among first-line patients in our clinical practice, and this is supported by the test and treatment strategy implemented nationwide [18,41].

Regarding virological response to first-line TLD, very high rates of viral suppression (>95%) were observed in real-life clinical settings after 14 months of TLD uptake, even with prior exposure to TLE. It is important to note that for homogeneity in the trend of data and for an optimal clinical relevance, data on viral load have been harmonized to showcase the rate of detectable VL before transition, which dropped down to 5.04% (at the threshold of >400 copies/mL) after 14 months of transitioning to TLD; this is in line with recent findings on TLD effectiveness worldwide [10,12,13,14,15,16,17,24,42,43,44,45,46,47]. TLE and TLD are once-daily fixed-dose combinations; hence, among similar adherence features, the major difference relies essentially on the regimen potency/genetic barrier likely driven by the DTG. Furthermore, with a median duration of 14 months and a viral nonsuppression rate of approximately 3%, it would be commendable to ensure a long-term monitoring of viremic patients in order to determine cases of infrequent nonadherence from cases of failure due to emerging DRMs to TLD, while emphasizing pharmacovigilance for an efficient patient outcome in the long run [48,49]. Importantly, neither age, gender, patient clinical status, nor ART duration were found to be associated with the viral suppression in these patients, indicating that the good virological response obtained is the primary outcome of an effective ART regimen, due essentially to the absence of preexisting DTG-resistance and possibly the scarcity of NRTI-DRMs in the current ART combination (indeed, exposure to TLE was relatively short to enable emergence or archiving of NRTI drug resistance). This finding is reassuring, encouraging a timely switch from TLE to TLD and warranting the extension of DTG-containing regimens even for pediatric populations in RLS just as recently reported [24,50]. Additionally, the few cases of nonsuppression were seemingly related to recent poor treatment adherence. Notably, recent cohorts in RLS reported the high acceptability and/or adherence to DTG-based regimens to be associated with the high rates of viral suppression observed with TLD-based first-line ART [43,44]. These findings suggest that enhanced therapeutic education and psychosocial support are key for a rapid decay of the viremia among patients initiating TLD-based first-line ART. Thus, this study brings additional emphasis on the high potency (i.e., high viral inhibitory capacity) of DTG observed within real-life clinical settings.

Moreover, no integrase resistance-associated mutations among virologically unsuppressed patients were reported when using a highly sensitive integrase-genotyping protocol that was validated locally [29]. The absence of INSTI resistance therefore confirms the high genetic barrier of DTG while encouraging the use of the current genotyping approach. The predominance of CRF02_AG subtype found in this study is in agreement with several other local studies [29,51,52,53,54,55,56], thus confirming the reliability from genotyping.

Although these findings are overall very encouraging, there is a need to continue monitoring the emergence of DRMs in this population, taking into consideration the suboptimal viral load coverage and risk of drug stock outs in RLS [9,11,49,53,57]. Moreover, pharmacovigilance of TLD remains a key component with major gaps in the current programmatic era, to ensure long-term safety alongside the reported efficacy of TLD in RLS [58,59,60].

The limited amount of data on the WHO clinical stages at treatment initiation might have led to some information bias. Nonetheless, the very high rates of clinical stages groups 1 and 2 suggest that this variable might not change even with full data coverage. Furthermore, as per national guidelines, viral load and NRTI resistance were not performed at treatment initiation, which could have helped to mitigate potential effects of baseline viremia and/or preexisting drug resistance mutations on the current TLD efficacy in both groups. This study was conducted during the COVID-19 pandemic, leading to some cases of exclusion due to limited clinic attendance for ART monitoring; however, the sample population was statistically sufficient for assessing the study primary outcomes.

## 5. Conclusions

In a nutshell, viral suppression is very high under first-line treatment after approximately 14 months following TLD initiation, even among patients with prior exposure to TLE in Cameroon. Our finding suggests that previous exposure to TLE is seemingly not a counterindication for transitioning from TLE to TLD as a public health approach in RLS. Interestingly, no independent factor was found to be associated with these high rates of virological control, suggesting that this observation is exclusively related to the strong pharmacological potency of dolutegravir (DTG). The current transition plan from TLE to TLD in similar African settings should therefore be encouraged, pending long-term ART monitoring for efficacy and drug-resistance surveillance, as well as safety monitoring through pharmacovigilance.

## Figures and Tables

**Figure 1 viruses-15-00018-f001:**
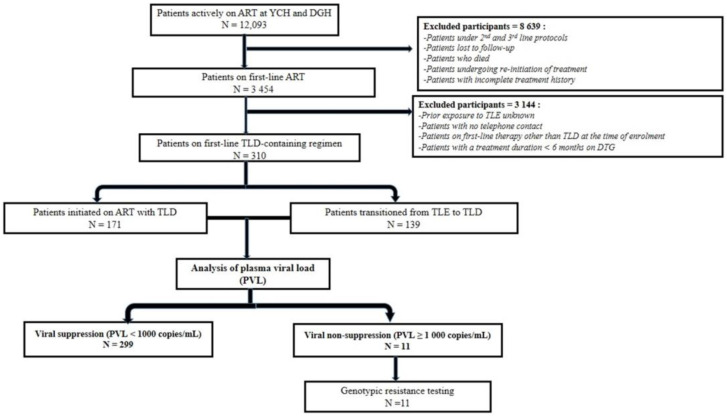
Flow chart of patients’ enrolment and follow-up in the study population. Legend. Bold represents subtitles.

**Figure 2 viruses-15-00018-f002:**
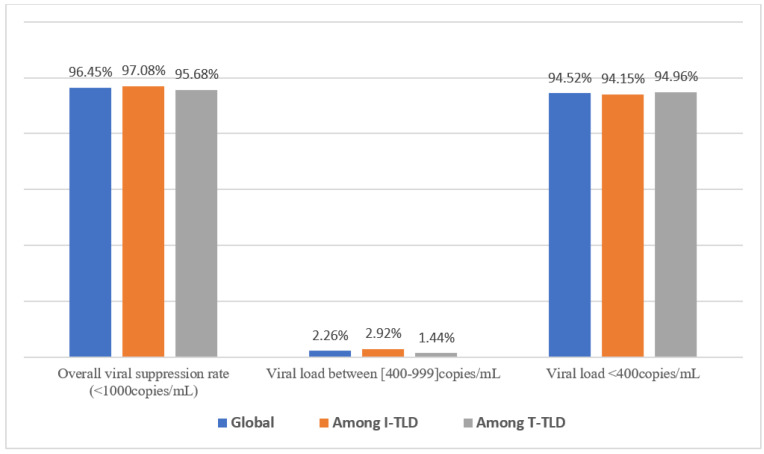
Virological response under TLD in first-line treatment among participants.

**Table 1 viruses-15-00018-t001:** Baseline characteristics of study participants.

Characteristics	Overall N = 310	I-TLD N = 171	T-TLD N = 139
Female	162	87	75
Male	148	84	64
Median age in years	41 (34–49)	39 (32–48)	42 (36–49)
Region of residence:			
Yaoundé	209	128	81
Douala	101	43	58
WHO clinical stage (n = 178) *:			
I	120	84	36
II	28	21	07
III	18	11	07
IV	12	07	05
Median ART-duration on first-line [IQR] (*months*)	19 (13–27)	14 (12–17)	28 (24.5–31)
Median ART-duration on TLD [IQR] *(months)*	14 (11–15)	14 (12–17)	14 (9–15)
Median viremia before transition [IQR] *(copies/mL)*	3258 (831–59,276)	/	3258 (831–59,276)

Legend: I-TLD refers to participants initiated on ART with TLD; T-TLD refers to participants transitioned from TLE to TLD; IQR refers to interquartile range. * Data regarding the WHO clinical stages at treatment initiation was available only for 178 participants.

**Table 2 viruses-15-00018-t002:** Evaluation of sociodemographic factors potentially associated with virological suppression in Cameroonian individuals treated with tenofovir + lamivudine + dolutegravir (TLD).

Variables	Virological Suppression on TLD		Odd Ratio [IC = 95%] (*p*-Value)	* Adjusted Odd Ratio [IC = 95%] (*p*-Value)
Age	Yes	No		
≤41 yrs	150	9	0.26 [0.05–1.22] (0.06)	1.64 (0.45–6.00) (0.45)
>41 yrs	129	2		
Sexe				
Female	159	3	3.03 [0.79–11.63] (0.08)	0.44 (0.12–1.63) (0.22)
Male	140	8		
Region of residence				
Douala	100	1	5.02 [0.63–39.81] (0.07)	1.27 (0.30–5.34) (0.74)
Yaoundé	199	10		
WHO stage				
I and II III and IV	139	29	0.53 [0.06–4.37] (0.47)	/
	29	1		
Duration on TLD				
≤14 months	179	6	1.24 [0.37–4.17] (0.76)	/
>14 months	120	5		

Legend: * Adjustment was made through multivariable analysis.

## Data Availability

Not applicable.

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
