# Peer review of "Dolutegravir-Based Regimen Ensures High Virological Success despite Prior Exposure to Efavirenz-Based First-LINE ART in Cameroon: An Evidence of a Successful Transition Model"

_viruses, 2022, doi:10.3390/v15010018_

Round 1
Reviewer 1 Report
Review for “Dolutegravir-based Regimen Ensures High Virological Success despite prior Exposure to Efavirenz-based First-Line ART in Cameroon: An Evidence of a Successful Transition Model”
This is a very well and clearly written study. However, there are some comments and clarifications I´d request to improve the understanding of the results.
General comments and questions:
Throughout the text the authors say they compared first-line virological response. However, in case of -T-TLD it is the TLD regimen is not first line anymore because they switched from TLE. I think this needs some rephrasing at several parts of the text.
Introduction:
Line 79-82: However, despite the predicted high effectiveness of TLD, the effect of previous exposure to a same backbone (TDF+3TC) gives room for possible existing drug resistant mutants in the current transition from TLE to TLD.
The part gives room for possible existing drug resistant mutants in the current transition from TLE to TLD is very general spoken which mutations exactly is referred to?
Methods:
If I understand it right viral suppression is defined as VL <1000 copies/ml. If that’s correct please include the definition into the methods.
At what point in time was the viral load testing performed? In the discussion the authors say viral suppression after 14 months. Was this the study protocol VL testing after 14 months? Please include the information on VL testing into the methods and also results section to clarify.
The age is at what reference date? I´d suggest to include this information into the
Statistical analysis
I believe you did a multivariable regression not a multivariate regression. Please check the definition of multivariable vs. multivariate regression and correct accordingly.
Why did you choose a p-value of 0.2 for the regression? If you did then the results in table 2a would be significant?
Results:
“The most represented age-group was [35 – 44] years (34.19%), followed by [25 – 34] years (21.29%), [45 – 54] years (20.32%), [55 – 64] years (10.65%), [15-24] years (4.52%), [65 – 74] years (2.26%), and finally that of [75 – 85] years (0.32%).”
This sentence could be presented as histogram.
Line 170 and table 1: When you refer to the WHO clinical stages I´d suggest to include the information again that it was the WHO clinical stage at treatment initiation for clarification.
Line 185: A space is missing.
Line 193: A space is missing.
“Of note, according to PVL threshold in I-TLD vs. T-TLD respectively, virological response was 2.92% (5/171) vs. 4.32% (6/139) at ≥1000 copies/ml (p=0.55); 2.92% (5/171) vs. 1.44% (2/139) at 400-999 copies/ml (p=0.36) and 94.15% (161/171) vs. 94.96% (132/139) at <400 copies/ml (p=0.55).”
This information could be included into figure 2. Maybe change to a histogram or consider another way of presenting the information into figure 2. Otherwise figure 2 doesn´t contain much information whereas this sentence is overloaded.
Line 202: See comment on multivariate regression above.
Table 2a: See question and comment on p-value of 0.2 above and clarify with regard to p-values for age, sex, residence.
Table 2a and table 2b could be combined into one table when you just add 3 more rows fo the adjusted odd ratios (aOR), the 95% CI and the p-values from table 2b.
Regression analysis: With few patients in the study and missing values for WHO clinical stages you could think about not excluding the patients with missing information but add the group missing. This might at least increase the power for the other variables and the regression analysis in general.
Discussion:
Line 217 ff: What is the situation in Cameroon with regard to the UNAIDS targets? Please include this information if available and discuss shortly.
Line 221: A space is missing.
“However, despite the tremendous successes of ART scale-up it is unlikely that there have been an increase in levels of pretreatment HIVDR among patients initiating ART [9,25].”
Did the authors mean likely or unlikely in the former sentence?
Line 22): “Participants to this study were aged 41 years in average and with a slight predominance (even though not significant) of females, …”
I´d suggest to phrase it like in the results where the authors said: There was a nearly equal distribution of males and female participants. Because 52.26% of females is rather equal.
Line 232 ff and Line 272 ff: I´d include the part on the WHO clinical stage and the limited data on it in the part at Line 232 ff and then mention it as a limitation shortly at Line 272 ff because otherwise the reader might wonder why you left this out of the discussion only to find it at the end.
Line 237: Include that it was in median 14 months of TLD uptake. Also see the comments and questions on the study protocol and VL testing.
Line 241: The authors speak of early ART failure. However, the median duration on TLD was 14 months. Please clarify what you mean by early failure.
Line 253: Where is this information. If the authors have the information maybe it’s worth including and presenting it? Also the information on reasons for switching could be useful.
Line 261: “Moreover, no integrase-resistance associated mutations among virologically unsup-260 pressed patients, …” I believe there the part was found is missing in the sentence.
Line 274-275: “Furthermore, as per national 274 guidelines, viral load and NRTI resistance were not done at treatment initiation,…” When was VL testing performed? See questions and comments above and clarify and include the information.
Conclusions:
Line 282-283: When was VL testing performed? See questions and comments above and clarify and include the information.
Author Response
Comments and Suggestions for Authors
Comment 1
Review for “Dolutegravir-based Regimen Ensures High Virological Success despite prior Exposure to Efavirenz-based First-Line ART in Cameroon: An Evidence of a Successful Transition Model”
This is a very well and clearly written study. However, there are some comments and clarifications I´d request to improve the understanding of the results.
General comments and questions:
Throughout the text the authors say they compared first-line virological response. However, in case of -T-TLD it is the TLD regimen is not first line anymore because they switched from TLE. I think this needs some rephrasing at several parts of the text.
Response 1: We thank the reviewer for this comment. However, TLD in Cameroon is officially the preferred first-line regimen; and as such, transition from TLE to TLD is not considered as a switch, but rather as a substitution in a patient not classified as ART failure according to national guidelines.
Comment 2
Introduction:
Line 79-82: However, despite the predicted high effectiveness of TLD, the effect of previous exposure to a same backbone (TDF+3TC) gives room for possible existing drug resistant mutants in the current transition from TLE to TLD. The part gives room for possible existing drug resistant mutants in the current transition from TLE to TLD is very general spoken which mutations exactly is referred to?
Response 2: We thank the reviewer for this comment. For specificity and clarity to readers, we have rephrased the sentence as follows: “However, despite the predicted high effectiveness of TLD, the effect of previous exposure to a same backbone (TDF+3TC) gives room for possible existing drug resistant mutants in the current transition from TLE to TLD (specifically with the K65R and M184V mutations known as major DRMs to TDF and 3TC/FTC respectively)” (see lines 79-89; page 2).
Comment 3
Methods:
If I understand it right viral suppression is defined as VL <1000 copies/ml. If that’s correct please include the definition into the methods.
Response 3: We thank the reviewer for this comment. We have revised accordingly and added a supporting reference (see line 133-134, page 3; and reference N°18).
Comment 4
At what point in time was the viral load testing performed? In the discussion the authors say viral suppression after 14 months. Was this the study protocol VL testing after 14 months? Please include the information on VL testing into the methods and also results section to clarify.
Response 4: We thank the reviewer for this comment. At the moment of the study, the median duration on TLD was 14 months for the entire study population. To ensure data homogeneity and to limit possible bias in the primary outcome of our study, we perform VL testing to all study participants. For this reason, the observed result of VL clearly reflects the virological response following “14 months” of exposure to TLD. Thus, we have revised the manuscript throughout as suggested to better clarify this point.
Comment 5
The age is at what reference date? I´d suggest to include this information into the
Statistical analysis
I believe you did a multivariable regression not a multivariate regression. Please check the definition of multivariable vs. multivariate regression and correct accordingly.
Response 5: We thank the reviewer for this observation. It is indeed “multivariable” instead of “multivariate”. We have revised accordingly (see line 166; page 3).
Comment 6
Why did you choose a p-value of 0.2 for the regression? If you did then the results in table 2a would be significant?
Response 6: We thank the reviewer for this comment. In fact, the threshold for significance is effectively p=0.05. However, from bivariate analysis, any p-value <0.2 could become significant after multivariable analysis (i.e. following adjustment of potential bias). Thus, the choice of a p-value of 0.2 for inclusion into the regression analysis ensures an optimal statistical robustness in the data interpretation. See reference for full description and understanding of this statistical approach (https://web.stanford.edu/~hastie/StatLearnSparsity/).
Comment 7
Results:
“The most represented age-group was [35 – 44] years (34.19%), followed by [25 – 34] years (21.29%), [45 – 54] years (20.32%), [55 – 64] years (10.65%), [15-24] years (4.52%), [65 – 74] years (2.26%), and finally that of [75 – 85] years (0.32%).” This sentence could be presented as histogram.
Response 7: We thank the reviewer for this suggestion. The figure has been provided as recommended (see list of figures now referring to figure x in the revised manuscript).
Comment 8
Line 170 and table 1: When you refer to the WHO clinical stages I´d suggest to include the information again that it was the WHO clinical stage at treatment initiation for clarification.
Response 8: We thank the reviewer for this suggestion. We have revised accordingly as suggested throughout the text.
Comment 9
Line 185: A space is missing.
Line 193: A space is missing.
“Of note, according to PVL threshold in I-TLD vs. T-TLD respectively, virological response was 2.92% (5/171) vs. 4.32% (6/139) at ≥1000 copies/ml (p=0.55); 2.92% (5/171) vs. 1.44% (2/139) at 400-999 copies/ml (p=0.36) and 94.15% (161/171) vs. 94.96% (132/139) at <400 copies/ml (p=0.55).”
This information could be included into figure 2. Maybe change to a histogram or consider another way of presenting the information into figure 2. Otherwise figure 2 doesn´t contain much information whereas this sentence is overloaded.
Response 9: We thank the reviewer for this suggestion. The figure has been structured to accommodate the data as indicated by the reviewer. This revised version also gives readers an easy understanding of the trend across VL ranges.
Comment 10
Line 202: See comment on multivariate regression above.
Response 10: we thank the reviewer for this comment. See detailed response to this comment provided above at response 5.
Comment 11
Table 2a: See question and comment on p-value of 0.2 above and clarify with regard to p-values for age, sex, residence.
Response 11: We thank the reviewer for the comment. Clarity to this comment has been aforementioned at response 6.
Comment 12
Table 2a and table 2b could be combined into one table when you just add 3 more rows for the adjusted odd ratios (aOR), the 95% CI and the p-values from table 2b.
Response 12: We thank the reviewer for the comment. The table has been revised accordingly and this gives a better readability of the paper.
Comment 13
Regression analysis: With few patients in the study and missing values for WHO clinical stages you could think about not excluding the patients with missing information but add the group missing. This might at least increase the power for the other variables and the regression analysis in general.
Response 13: We thank the reviewer for the comment. The group on missing data has been added as suggested.
Comment 14
Discussion:
Line 217 ff: What is the situation in Cameroon with regard to the UNAIDS targets? Please include this information if available and discuss shortly.
Response 14: We thank the reviewer for this comment. We included the information as suggested (see lines 237-239 and reference N°33; page7).
Comment 15
Line 221: A space is missing.
“However, despite the tremendous successes of ART scale-up it is unlikely that there have been an increase in levels of pretreatment HIVDR among patients initiating ART [9,25].” Did the authors mean likely or unlikely in the former sentence?
Response 15: We thank the reviewer for this observation. The term “unlikely” seem not to convey the idea we wanted to communicate; we therefore rephrased as follows: “However, despite the tremendous successes of ART scale-up it is very unfortunate that there have been an increase in levels of pretreatment HIVDR among patients initiating ART” (see line 241; page 7).
Comment 16
Line 22): “Participants to this study were aged 41 years in average and with a slight predominance (even though not significant) of females, …” I´d suggest to phrase it like in the results where the authors said: There was a nearly equal distribution of males and female participants. Because 52.26% of females is rather equal.
Response 16: We thank the reviewer for this suggestion. We have rephrased accordingly as suggested (see lines 249-250; page 7).
Comment 17
Line 232 ff and Line 272 ff: I´d include the part on the WHO clinical stage and the limited data on it in the part at Line 232 ff and then mention it as a limitation shortly at Line 272 ff because otherwise the reader might wonder why you left this out of the discussion only to find it at the end.
Response 17: We thank the reviewer for this comment. We have revised accordingly (see lines 253-255; page 8).
Comment 18
Line 237: Include that it was in median 14 months of TLD uptake. Also see the comments and questions on the study protocol and VL testing.
Response 18: We thank the reviewer for this comment. We have revised accordingly throughout.
Comment 19
Line 241: The authors speak of early ART failure. However, the median duration on TLD was 14 months. Please clarify what you mean by early failure.
Response 19: We thank the reviewer for this comment. We have clarified the statement as follows: “Furthermore, with a median duration of 14 months and a viral non-suppression rate around 3%, it would be commendable to ensure a long-term monitoring of viremic patients in order to distinct cases of infrequent non-adherence from cases of failure due to emerging DRMs to TLD, while emphasizing on pharmacovigilance for an efficient patient outcome in a long run” (see lines 258-260; page 7).
Comment 20
Line 253: Where is this information. If the authors have the information maybe it’s worth including and presenting it? Also the information on reasons for switching could be useful.
Response 20: We thank the reviewer for this comment. This has been addressed accordingly.
Comment 21
Line 261: “Moreover, no integrase-resistance associated mutations among virologically unsup-260 pressed patients, …” I believe there the part was found is missing in the sentence.
Response 21: We agree with the reviewer’s comment. It has been addressed accordingly.
Comment 22
Line 274-275: “Furthermore, as per national 274 guidelines, viral load and NRTI resistance were not done at treatment initiation,…” When was VL testing performed? See questions and comments above and clarify and include the information.
Conclusions:
Line 282-283: When was VL testing performed? See questions and comments above and clarify and include the information.
Response 22: We thank the reviewer for this comments. In effect, as early mentioned, VL testing was performed at the moment of enrolment into the current study. The “14 months” here refers to the median duration on TLD.
Reviewer 2 Report
Authors present a strong introduction for the clinical state of ART in third world countries and potential for resistance, thus warranting the current study. Study design is sound and well described. Results are clearly presented. Overall sound report.
Only question is whether the subjects were followed at more frequent intervals to allow assessment for kinetics of viral suppression between the 2 groups.
Author Response
Comment 1
Authors present a strong introduction for the clinical state of ART in third world countries and potential for resistance, thus warranting the current study. Study design is sound and well described. Results are clearly presented. Overall sound report.
Only question is whether the subjects were followed at more frequent intervals to allow assessment for kinetics of viral suppression between the 2 groups.
Response 1: We thank the reviewer for this question. Most participants (especially in the I-TLD arm) did not have many records of viral load testing to allow assessment for viral suppression kinetics, as they were all initiating therapy. Even though patients on T-TLD (transitioning from TLE to TLD) had been on treatment for a longer period, we could not compare viral load kinetics between the two arms.
Reviewer 3 Report
The authors aimed to compare virological response of antiretroviral treatment consisting of tenofovir+lamivudine+dolutegravir according to prior exposure to tenofovir+lamivudine+efavirenz. The authors observed very high rates of viral suppression after 14 months of TLD-uptake, in the group of patients with and without prior exposure to TLE. Although the manuscript is interesting, I have a few suggestions on how it might be further improved:
1) I believe that at the end of the introduction section, the authors could describe better the reasons which led them to conduct this study.
2) The authors have used chi square test to describe the associations between variables. However, the chi square test is invalid in case of having fewer than 5 observations. In those cases the authors should use another test, for example Fisher exact test.
3) The authors declare that 89.93% of T-TLD had a detectable viral load before the transition. Why are these numbers so huge? Is it due to drug resistance or the lack of adherence or another reasons? I think that it should be discussed in the discussion.
4) In materials and methods section and also in Table 1 T-TLD refers to participants transitioned from TLE or TLN to TLD. However, in line 181 authors declare that there were no participants transitioning from TLN, so maybe it would be clearer to just describe T-TLD as patients transitioned from TLE?
5) In Table 2a, p value is presented to the second decimal place and in Table 2b, to the fourth decimal place, I think it would be better to standarize it for example to the third decimal place.
Author Response
The authors aimed to compare virological response of antiretroviral treatment consisting of tenofovir+lamivudine+dolutegravir according to prior exposure to tenofovir+lamivudine+efavirenz. The authors observed very high rates of viral suppression after 14 months of TLD-uptake, in the group of patients with and without prior exposure to TLE. Although the manuscript is interesting, I have a few suggestions on how it might be further improved:
Comment 1) I believe that at the end of the introduction section, the authors could describe better the reasons which led them to conduct this study.
Response 1: We thank the reviewer for this comment. We have added some comments to further describe the reasons for this study within the Cameroonian context as requested (see lines 94-97; page 2).
Comment 2) The authors have used chi square test to describe the associations between variables. However, the chi square test is invalid in case of having fewer than 5 observations. In those cases the authors should use another test, for example Fisher exact test.
Response 2: We thank the reviewer for this comment. Indeed this was done during analyses and we revised the sentence as follows: “Chi square or Fisher’s exact test was used to describe the associations between variables wherever appropriate; with p-values <0.05 considered statistically significant; logistic regression was used for multivariate multivariable analyses for variables showing a p-values ≤ 0.2 from bi-variable analyses” (see lines 162-165; page 4).
Comment 3) The authors declare that 89.93% of T-TLD had a detectable viral load before the transition. Why are these numbers so huge? Is it due to drug resistance or the lack of adherence or another reasons? I think that it should be discussed in the discussion.
Response 3: We thank the reviewer for this observation. For homogeneity in the trend of data and for an optimal clinical relevance, we have harmonized this observation to showcase the rate of detectable VL at the threshold of >400 before transition (%) which dropped down to % after 14 months from transitioning to TLD. In this frame where TLE and TLD are once daily fixed dose combinations hence similar adherence features, the major difference relies essentially on the regimen potency/genetic barrier likely driven by the DTG. (see lines 252-255; page 7).
Comment 4) In materials and methods section and also in Table 1 T-TLD refers to participants transitioned from TLE or TLN to TLD. However, in line 181 authors declare that there were no participants transitioning from TLN, so maybe it would be clearer to just describe T-TLD as patients transitioned from TLE?
Response 4: We thank the reviewer for the suggestion. We have revised accordingly throughout the manuscript.
Comment 5) In Table 2a, p value is presented to the second decimal place and in Table 2b, to the fourth decimal place, I think it would be better to standardize it for example to the third decimal place.
Response 5: We thank the reviewer for this suggestion. We have revised and presented the values to the second decimal place throughout.
Round 2
Reviewer 3 Report
The manuscript was improved and I accept it in present form.